# 3D Sensitivity Zone Mapping in a Multi-Static, Microwave Breast Imaging Configuration

**DOI:** 10.3390/s25165150

**Published:** 2025-08-19

**Authors:** Paul Meaney, Zamzam Kordiboroujeni, Keith Paulsen

**Affiliations:** Thayer School of Engineering, Dartmouth College, Hanover, NH 03755, USA

**Keywords:** sensitivity zone, microwave imaging, breast, three dimensional (3D), reciprocity, monopole antennas

## Abstract

One of the keys to medical microwave tomography is understanding the sensitivity of transmit–receive signals to changes in the electromagnetic properties to be reconstructed. This information is embedded in the Jacobian matrix for traditional inverse problem approaches and is a function of transmitter–receiver design characteristics and associated signal radiation/detection patterns. Previous efforts focused primarily on the 2D imaging problem for which sensitivity maps were generated in a single plane. In this paper, we describe sensitivity maps for the full 3D problem for monopole transceivers and their implications for associated antenna array configurations, including imaging zone coverage and computational efficiency.

## 1. Introduction

Antenna design for a near field, multi-static imaging system has important similarities and differences relative to classical electromagnetic systems such as radars and communication systems [1,2]. Of primary importance is the detection of transmitted signals by receivers [3], which must account for factors such as transmitted power, operating frequency, transmitter–receiver separation, transmission medium, measurement dynamic range and transmitter/receiver size [4,5]. For imaging systems, equally important is signal coverage (of the target), multi-path signals and transmitter–receiver orientation [6,7]. In the case of imaging human tissue, signal attenuation is often substantial in terms of signal reflection, refraction and conductive losses [8]. Historically, vector network analyzer (VNA) dynamic ranges were limited to 100 dB or so, which placed a premium on using directive antennas to ensure sufficient signal-to-noise ratio (SNR) [9,10,11]. Here, setting a noise floor lower than −100 dBm required reductions in measurement bandwidths to levels where acquisition times were excessive, and not practical for real-time applications like radar and telecommunications [3]. The net implication was the need for larger antennas, which constrained array configurations when multiple antennas were arranged around the target [12,13].

While the basic premise of being able to measure the signal is still critical, the advent of VNAs with dynamic ranges exceeding 140 dB has made the classic imperative for high dynamic range less pivotal [3]. As a result, we have become progressively more interested in the sensitivity of measured signals to dielectric property variations within the target region [14,15]. Prior work demonstrated that these sensitivity maps can be extracted from rows of the Jacobian matrix used in the Gauss-Newton iterative reconstruction process [14], which is the multi-variate extension of Newton’s Method for determining a single root of a known function [16]. The latter essentially finds the value at a new iteration by subtracting the function divided by its derivative at the previous iteration from the associated current value [17]. Extension to the multi-variate problem involves an expansion in the number of variables and measurement observations, along with inflation of associated derivatives to a matrix (called the Jacobian) of partial derivatives defined by the number of observations and variables [17]. This multi-variate expansion is used widely in imaging and is often expanded from real-valued to complex-valued formulations [18]. In the electromagnetic case, the Helmholtz equation is often used as the governing function [19].

We previously presented the 2D case and showed that the sensitivity was high in cigar-shaped concentrations between antenna pairs, but relatively low elsewhere. In this paper, we extend the analysis to the 3D case for a monopole antenna array circumscribing a target (anatomical) region of interest, and focus on two vertical and orthogonal planes—the first transecting both vertical transmitting and receiving monopole antennas and the second oriented perpendicularly at the center point between antennas. Critical to the assessment is whether property changes outside of the high-sensitivity zones impact the measurements, in particular, in areas above/below the active segments of the antenna elements and in zones at interfaces with the body (i.e., the target).

Generating sensitivity maps is not necessarily an explicit means for improving an imaging approach. However, these maps are influenced by the design of key elements of the system. For instance, while not optimal in a classical sense of radiation efficiency and directivity (desirable characteristics generally associated with communications and radar systems), monopole antenna arrays produce sensitivity maps with characteristics that are attractive for certain types of imaging applications [14]. Thus, the sensitivity analysis tool can provide valuable insight into the design of features that are most critical in a near-field imaging system [20].

Multiple factors influence characteristics of the sensitivity maps, in particular antenna design and operating frequency. For the monopole, a key factor is antenna length. Equally important, sensitivity maps change substantially with frequency. In the cases evaluated here, monopoles are loaded resistively by a lossy coupling bath and typically operate over an extended bandwidth—in our hands, the range is approximately 500 MHz to 3 GHz. In this paper, we evaluate the effects of antenna length and frequency to infer optimal settings for our imaging system.

Our goal in this paper is to provide a complete description of 3D sensitivity analysis and its use as a design technique for examining the implications of antenna choice and overall configuration. As part of the Results section, we demonstrate an experimental imaging case where shorter antenna lengths contributed to increased artifacts near the target/background interface. Differences in sensitivity maps for different length antennas were subtle but confirmed the observed experimental effects. The technique has also provided insight into realistic antenna array configurations by incorporating external features such as feedlines and support structures, the effects of which have mostly been ignored in the literature.

The Methods section describes the sensitivity maps, including a discussion of how the adjoint technique is used to accelerate computation of the Jacobian matrix [14,15]. The Results section provides representative maps of sensitivity zones for a number of antenna pairs. Maps are also computed for variation in antenna length and operating frequency. An overall goal is target coverage (e.g., breast), which necessitates imaging as close to the tissue interface as possible (i.e., chestwall), placing a premium on signal sensitivity in this region.

## 2. Methods

### 2.1. Expansion of Newton’s Method to the Multi-Variate Case

In its simplest representation, the Gauss–Newton technique derives directly from a multi-variate expansion of Newton’s method for finding roots of an equation with a single independent variable. The solution is obtained iteratively, where the estimated value at a new iteration is determined by the sum of the previous value minus the function divided by its derivative at the previous value. Usually, this process is repeated until the function is sufficiently close to a desired value, or the difference in successive iterates approaches zero [16]:(1)xn+1= xn− fxnf′xn
where *x_n_* and *x_n_*_+1_ are the variables at iterations n and n + 1, respectively, and f*(•)* and *f′(•)* are the function and its first derivative, respectively. Extension to multiple variables expands the iterative solution process to a system of equations involving a matrix of partial derivatives [17]:(2)$$\left[J\right]\;\left\{∆x\right\}=\;\left\{y-f\left(x\right)\right\}$$
where ∆x is the vector of variable changes *(x_n_ − x_n_*_−1_*)*, *y* and *f(x)* are the vectors of observations and the function, respectively, and *J* is the Jacobian matrix. The column dimension of *J* is the number of observations, while the associated row dimension is the number of variables. *Δx* is simply added to the previous vector, *x_n_*_−1_, to estimate the new vector, *x_n_*.

### 2.2. Sensitivity Maps

Many parameter estimation problems are implemented in terms of only real-valued functions. However, expansion to complex-valued variables is relatively straightforward [18]. Microwave tomography is a classic example and utilizes the Helmholtz equation as the governing equation, where the number of property variables to be estimated is doubled. Correspondingly, measurements are complex functions and often represented in polar form as their log magnitude and phase. Here, the number of observations is similarly doubled [15]. To simplify the formulation, these two-by-two factors are condensed into single 2 × 2 submatrices so that the overall formulation can still be cast in terms of the respective number of variables and observations [15]. While common to reconstruct these problems in terms of their real and imaginary constituents, numerous transformations have been implemented to improve the robustness of the recovery process [21]. One common approach is the log transformation, where the electromagnetic fields are expressed in terms of their log magnitude and phase [22].

An informative way to visualize sensitivity to property change is to map the Jacobian matrix used in microwave tomography to generate property updates during iterative image formation with, for example, Gauss–Newton methods. Some imaging techniques do not form a Jacobian matrix explicitly, but have developed algorithms for essentially constructing the equivalent. Cui et al. [23] implemented an integral formulation to construct what is effectively a Jacobian matrix that was later used by Shea et al. [24] and Karadima et al. [25]. Van den Berg and Kleinman [26] realized a contrast source inversion approach that computed the associated gradients via Frechet derivatives. Gilmore et al. [10] described a similar approach except that they added a multiplicative regularization.

For our system, we utilize an array of 16 monopole antennas configured on a 15.2 cm diameter circle, as shown in Figure 1. In general terms, the Jacobian can be formed as a two-dimensional matrix where the number of columns corresponds to the number of pixels (n_p_) in an imaging zone and the number of rows corresponds to the number of measurements—in our case, the number of transmitters (n_t_) x the number of receivers (n_r_) per each transmitter (Figure 2). In this case, n_t_ x n_r_ is the total number of observations. Each term can be written as ∂Et,r∂εp, which represents change in the electric field (*E*) due to a signal transmitted from t and received at r with respect to an infinitesimal change in dielectric properties (ε) at pixel p. If one examines a single row in the resulting matrix, it contains derivative values at all pixels for a single transmitter and receiver, and can be considered a map of the sensitivity within the associated imaging zone.

Here, options exist for expressing sensitivity because both *E* and ε are complex; thus, each matrix element is actually a 2 × 2 submatrix of real-valued entries. Similarly, effects of dielectric properties can be expressed in alternative forms, for example, as the complex relative permittivity $\varepsilon_{r}+j\varepsilon_{i}$ or the complex wave number squared − $k_{r}^{2}+jk_{i}^{2}$. For the case where we utilize the wave number squared, the four partial derivative terms of the submatrix are(3)$$\frac{\partial E_{R}}{\partial k_{r}^{2}}\;\;\;\frac{\partial E_{R}}{\partial k_{i}^{2}}$$(4)$$\frac{\partial E_{I}}{\partial k_{r}^{2}}\;\;\;\frac{\partial E_{I}}{\partial k_{i}^{2}}$$
where $E_{R}$ and $E_{I}$ are the real and imaginary parts of the electric fields, respectively.

In addition, we have previously utilized a log transformation for our imaging algorithm, which requires minor modifications (utilization of the quotient rule) to the Jacobian matrix derivation [20]:(5)$$\frac{\partial \Gamma }{\partial k_{r}^{2}}=\;\frac{E_{R}\frac{\partial E_{R}}{\partial k_{r}^{2}}+\;E_{I}\frac{\partial E_{I}}{\partial k_{r}^{2}}}{E_{R}^{2}+\;E_{I}^{2}}\;\;\frac{\partial \Gamma }{\partial k_{i}^{2}}=\;\frac{E_{R}\frac{\partial E_{R}}{\partial k_{i}^{2}}+\;E_{I}\frac{\partial E_{I}}{\partial k_{i}^{2}}}{E_{R}^{2}+\;E_{I}^{2}}$$(6)$$\frac{\partial }{\partial k_{r}^{2}}=\frac{E_{I}\frac{\partial E_{R}}{\partial k_{r}^{2}}+E_{R}\frac{\partial E_{I}}{\partial k_{r}^{2}}}{E_{R}^{2}+E_{I}^{2}}\;\;\frac{\partial }{\partial k_{i}^{2}}=\frac{E_{I}\frac{\partial E_{R}}{\partial k_{i}^{2}}+E_{R}\frac{\partial E_{I}}{\partial k_{i}^{2}}}{E_{R}^{2}+E_{I}^{2}}$$
where *Γ* and *Φ* are the log magnitude and phase, respectively. While we have previously presented examples of these sensitivity maps from a 2D imaging algorithm [14], they are generalizable to 3D forms.

Further expansion is possible and has been implemented to reconstruct images from multi-frequency data [24,27,28]. In these cases, dielectric properties are represented as a dispersion function, where the algorithm then estimates the coefficients of the function instead of the actual properties [24,29]. The advantage here is that the approach allows the algorithm to access additional observational data, which is beneficial because these problems are generally ill-posed [24,30].

Historically, for the finite element-based (FE) approach introduced by our group, the Jacobian matrix was generated through a series of matrix decompositions with multiple back substitutions of the system used in computing electric field distributions [31], which was a computationally intensive process, similarly to competing approaches by Cui et al. [23] and others. Advances by Fang et al. demonstrated that when utilizing the Adjoint Method [15] (which invokes reciprocity theory), matrix elements could be formed by inner products of field distributions for transmission at antennas t and r, along with multiplication by an appropriate diagonal weighting. More recently, Hosseinzadegan et al. [14] showed that each row of the Jacobian matrix could be formed by multiplying the same field distributions together with a similar weighting matrix, which reduced the computational effort to order N (vector-vector dot product) for an N-dimensional system. The innovation was implemented in 2D and resulted in a Jacobian matrix computation time of ~5 msecs on a laptop computer [14], essentially eliminating the computational burden of Jacobian matrix construction.

This description is somewhat simplified because the dielectric properties and electric fields are complex-valued, and the electric fields are also vector quantities. The reader is referred to Fang et al. [15] for complete derivations. Previously, we considered the 2D scalar electric field case (i.e., field orientation normal to the 2D plane); however, production of the orthogonal sensitivity maps for 3D cases is straightforward. In this paper, we examined cases where (a) the planes intersect both the transmitting and receiving antennas, and (b) the plane is centered between the antennas and is perpendicular to both the horizontal plane in (a) and the previously published results.

### 2.3. Vector–Vector Multiplication

Simulated fields were computed using CST Studio Suite (Dassault Systemes—Vélizy-Villacoublay, France). For each field solution, the excitation was initiated inside the coaxial cable, assuming a 50 ohm impedance. The distribution was based on a monopole antenna of prescribed lengths, which extends into a lossy homogeneous coupling fluid. The monopoles were extensions of coaxial cables—RG-402—with a center conductor diameter of 0.93 mm, and an outer conductor diameter of 3.0 mm. The insulator was polytetrafluoroethylene (PTFE). Solutions were then mapped to a uniform grid overlaying the geometry described in the model. The separation between each monopole was determined by its position on the 15.2 cm diameter circle, which defined the antenna array. Once values were computed at each grid point, they were multiplied together and their magnitude was computed and converted to decibels (dB) via X (dB) = 20 × log_10_(Y), where Y is the value prior to conversion and X is the value after conversion, respectively. Depending on the plane of interest, contour plots were computed and displayed using Matlab R2025a.

## 3. Results

### 3.1. Sensitivity Maps for 1300 MHz Fields in x–z, x–y and z–y Planes

For the purposes of presenting representative maps in 2D planes, the *y*-axis is oriented vertically and the x- and z-axes are in the horizontal plane. Figure 3 shows two views of an antenna pair with the horizontal presentation plane cutting across the antennas in the x–z plane. Figure 4a shows the magnitude of the 1.3 GHz electric field distribution in the x–z plane for transmitter 1, assuming 16 evenly spaced monopole antennas distributed on a 15.2 cm diameter circle and oriented in the y-direction—y is the vertical direction, while z and x axes are in the plane perpendicular to the antennas. The monopole antennas are 3.5 cm long and radiate into a homogeneous bath comprised of an 88% glycerin bath (er = 14.9, and sigma = 1.00 S/m at 1300 MHz). In this plane, the field distributions are isotropic around the monopole antennas, and their magnitudes decay substantially as a function of radius.

Figure 4b–d show the corresponding x–z sensitivity maps for three representative antenna pairs: 1–9, 1–7 and 1–5, respectively (antennas numbered consecutively from 1 to 16 around the imaging zone). Vertical height of the plane is the midpoint of the monopole antennas. The higher intensity zones are cigar-shaped regions that extend between representative antenna pairs and decay precipitously outside of these zones. The scaling is referenced to the magnitude in the center of the cigar shape, plus and minus 20 dB. Input signal levels for the two antennas were set to the same value for each antenna in each case, and were essentially subtracted out when converted to logarithmic format. Similarly, the constant used to convert the product of the two field distributions to sensitivity terms in the Jacobian matrix was also subtracted out. In this way, the center value of each scale changed with overall field strength—higher values when antennas were closer (Figure 4d) and lower values when further away (Figure 4b). While some saturation occurs on both ends of the range (outside of the enhanced sensitivity for the low end and directly around the antennas for the high end), the span highlights characteristics inside the band between antennas that are of most interest. While the beam pattern from antenna 1 radiates uniformly around itself, fields received at the remaining (other) antennas vary due to perturbations in the space directly between the transmitter (antenna 1) and receivers (other antennas). The intervening medium is lossy, and property changes in spaces outside of these elevated sensitivity zones have less impact on measurements obtained at the receiving antennas. Since the imaging zone was a cylinder inside the monopole antennas, as depicted in Figure 1, the elevated sensitivity zones were largely homogeneous over the imaging space.

Figure 5 shows two views of an antenna pair with the vertical presentation plane cutting through the antennas and feedlines in the x–y plane. Figure 6a shows the magnitude of the 1.3 GHz field distributions in the x–y plane for transmitter 1. In this view, the fields emanate horizontally with little contribution in the vertical direction. The overall beam pattern is effectively toroidal.

Figure 6b–d present sensitivity map magnitudes in the x–y plane for three representative antenna pairs: 1–9, 1–7 and 1–5, respectively, radiating at 1.3 GHz. The scaling is set similarly to Figure 4 in order to highlight characteristics with the bands between antennas. Here, the sensitivity zone is capped on the high end across a horizontal line passing just above the antenna tips, whereas the lower side appears like a broad inverted parabola in which the mid widths of the sensitivity zones are comparable to the antenna lengths and extend down the lengths of the associated coaxial feedlines. The top of the sensitivity zone is relatively flat for the 1–5 (shortest distance) antenna pair, but bows upwards more so for the two larger spaced antenna pairs: 1 and 7 and 1 and 9, respectively.

Similarly, since the imaging zone was a cylinder inside the monopole antennas, as depicted in Figure 1, the elevated sensitivity zones were mostly homogeneous over the imaging space.

Figure 7 shows two views of an antenna pair with the vertical presentation plane cutting across the axis between antennas in the y–z plane. Finally, Figure 8a–c show sensitivity maps for 1.3 GHz fields in the z–y plane transecting the center of the two associated monopole antennas for three representative pairs: 1–9, 1–7 and 1–5, respectively. The scaling is again similar to Figure 4. Here, the maximum sensitivity zone is circular with approximately 7–8 cm in diameter. This shape, when considered in conjunction with those in Figure 4 and Figure 6, indicates that the elevated sensitivity zones are essentially cylinders extending from one antenna to the other.

### 3.2. Variation with Frequency

More sensitivity variation occurs in the x–y planes, and Figure 9 explores these maps as functions of antenna operating frequency for the same three antenna pairs: 1–9, 1–7 and 1–5, respectively, with the same 3.5 cm monopole lengths and distribution scaling as in the prior figures. The upper part of the sensitivity zone tends to bow upwards with increasing frequency and larger antenna spacing. In clinical practice, we position antennas close to the chest wall to image as much of the breast volume as possible. Extending sensitivity regions upward may cause signal disruptions resulting from the air-coupling bath interface. However, we have been able to recover good images of the breast over multiple horizontal planes, including close to the chest wall. While the latter tend to have more artifacts, they are generally manageable, and we still recover usable images even when the tips of the antennas are positioned within 2 mm of the air-liquid interface.

### 3.3. Variation with Antenna Length

In recent phantom imaging experiments, we investigated the use of 2.5 cm long monopole antennas as part of our effort to combine microwave imaging with MRI [32] because a premium exists on reducing the overall height of the imaging tank to fit inside the MR bore during a patient imaging exam. Unfortunately, the shorter antennas contributed to a substantial increase in imaging artifacts (relative to the 3.5 cm long monopole antennas) as the array approached the interface at the top of the tank. Previous studies comparing the beam width and direction of the two antenna designs did not demonstrate significant differences arising from the two antenna lengths [15]. Here, we explored differences in their corresponding sensitivity maps as a way of confirming differences in the observed performances of the two antenna arrays during phantom imaging experiments.

Figure 10 shows the same sensitivity map magnitudes as Figure 9 for monopole antenna lengths of 2.5 cm. The tendency for the upper part of the sensitivity zone to bow upwards with increased frequency and larger antenna spacing appears to be slightly accentuated by the shorter (2.5 cm) antenna lengths. The elevated sensitivity bands also appear wider for the shorter antennas, for example, in the Antenna 1–7, 1.1 GHz sensitivity maps (Figure 9e vs. Figure 10e), which suggests that the longer antennas provide a more focused imaging plane.

Because the sensitivity maps for the 3.5 and 2.5 cm antennas are so similar, we computed differences in their sensitivity maps for one example (1300 MHz and antenna pair 1–9) and expanded the scale (Figure 11). Two features are noteworthy. First, the region around and just above the tips of the 2.5 cm antennas is accentuated compared with that for the 3.5 cm antennas, and increased monotonically with frequency. Given that we typically bring the antenna tips within 2 mm of the liquid/air interface, the proximity of this enhanced zone likely contributed to aberrations in the fields that subsequently led to image reconstruction artifacts. Second, in the space between the antennas and just above their tips, a significantly enhanced sensitivity zone occurred for the 2.5 cm antennas compared to that for the 3.5 cm antennas, which also increased with frequency. Differences are on the order of 1.0–2.0 dB extending more than 1 cm above the antenna tips, which would certainly cause additional signal interactions in the space above the air/liquid interface, and ultimately contribute to image artifacts.

## 4. Discussion

Sensitivity maps provide an additional tool for analyzing the impact of design parameters in a multi-channel, near-field imaging system. Here, we considered monopole antennas positioned in a circumferential imaging array. From one perspective, these transmitters might not be considered effective choices because they radiate isotropically, with much of their signal not being used to image the target. However, by examining their sensitivity maps, monopole antennas provide effective responsiveness with concomitant complete coverage of the target within the imaging zone when placed close to the target, which becomes much more challenging for larger and more directive antennas.

While monopoles are essentially isotropic in the horizontal plane (perpendicular to the axis of the antenna), their sensitivities are elevated in cigar-shaped zones that extend from one antenna to another, which is expected in a lossy medium because property perturbations outside the intervening space between transmit/receive antennas have little effect on the measurements. While these zones of elevated sensitivity do not individually cover the entire imaging region of interest (the target), collectively they do provide coverage. Thus, for larger imaging zones (e.g., when antennas are positioned on a larger diameter), sensitivity zone overlap is reduced, especially in more peripheral locations, which is likely to degrade image quality.

We argued previously that recovered microwave images are essentially weighted averages of properties within a 2–3 cm thick slab centered approximately at the center of the active part of the antenna [33]. The sensitivity plots reported here suggest that the center of the slab is ~0.5 cm above the step transition from the coaxial feedline to the monopole antenna. If the 3 dB beamwidth from the sensitivity value in the center is used to define an effective imaging thickness, it is approximately 4.4 cm, whereas extending the criterion to 6 dB expands the imaging thickness to be closer to 6.2 cm. The former is larger than estimated from our earlier observations, but is reasonably consistent with prior experience.

Magnitude plots of sensitivity show elevated values closer to individual antennas relative to the mid region between antennas in the highlighted sensitivity zones. We have noted in the past that our imaging technique recovers inclusion sizes and properties more accurately when they are physically closer to the antennas relative to being located more interiorly. Efforts to compensate for the decreasing sensitivity with depth have led to radially biased regularization procedures that progressively weight the interior more heavily [34]. Accordingly, more accurate reconstructions are possible in smaller imaging zones since inclusions are closer to associated antennas, as illustrated in Fang et al. [35].

While the sensitivity zones extend downward along antenna feedlines, the extension is not problematic for image reconstruction, at least in the breast, because the tissue is closest to the antennas at the level of their tips and tapers inward until approaching the nipple. Thus, the elevated sensitivity near the feedlines does not overlap with the desired imaging field of view and has little, if any, negative impact on recovered images.

Recovering images as close to the chest wall as possible is important for the detection of breast disease. The sensitivity maps reported here suggest minimal impact on the measurements from property variations in regions above the antenna tips, which indicates antennas can be positioned close to the body. The assumption is consistent with our earlier clinical results, which confirmed that data can be acquired and images reconstructed with minimal artifacts when antenna tips are positioned within 1–2 mm from the skin.

Sensitivity maps for the 2.5 cm antennas do have slightly enhanced values around and above the antenna tips relative to their 3.5 cm antenna counterparts, and the effect is increased markedly with frequency. An enhanced zone also exists in the difference maps between the antennas and extends slightly above the level of the tips, which also increases with increasing frequency. In a series of phantom imaging experiments, we found higher levels of artifacts in reconstructed images for cases where antenna tips were 1–2 mm below the liquid bath level for the 2.5 cm antennas relative to the 3.5 cm antennas. We determined previously that these differences in artifact levels were due to differences in antenna lengths, and the sensitivity map differences reported here, though subtle, appear to confirm our prior empirical findings.

The results presented here only consider the design parameters for a monopole antenna-based system. Given that the Jacobian row distributions are essentially multiplications of field patterns from associated antennas, they would look considerably different for more directive antennas—such as dielectrically loaded waveguide antennas [36]. For instance, the sensitivity maps would likely be different, especially for adjacent transmit and receive antennas (assuming they are configured on a circle on a larger diameter because the more directive antennas are larger and occupy more space). Here, the monopoles may have advantages, and the type of sensitivity map analysis presented would be valuable in a future investigation.

## 5. Conclusions

Bi-static transmission-mode sensitivity maps confirm the imaging zone is confined to a slab spanning the antenna tips to the shoulder transitions between feedlines and the monopole antennas, which is counterintuitive given monopole radiation patterns extend well beyond this region. While signal loss from the isotropic radiation pattern of monopole antennas is a common criticism, their sensitivity maps indicate the loss is readily compensated with high dynamic range, commercially available vector network analyzers which allow the advantages of monopole antenna to be exploited, including their placement of close to the target [35] and their accurate modeling within the overall reconstruction process [31]. The latter is readily implemented with a discrete dipole approximation which reduces computation time from minutes to a few seconds [14].

## Figures and Tables

**Figure 1 sensors-25-05150-f001:**
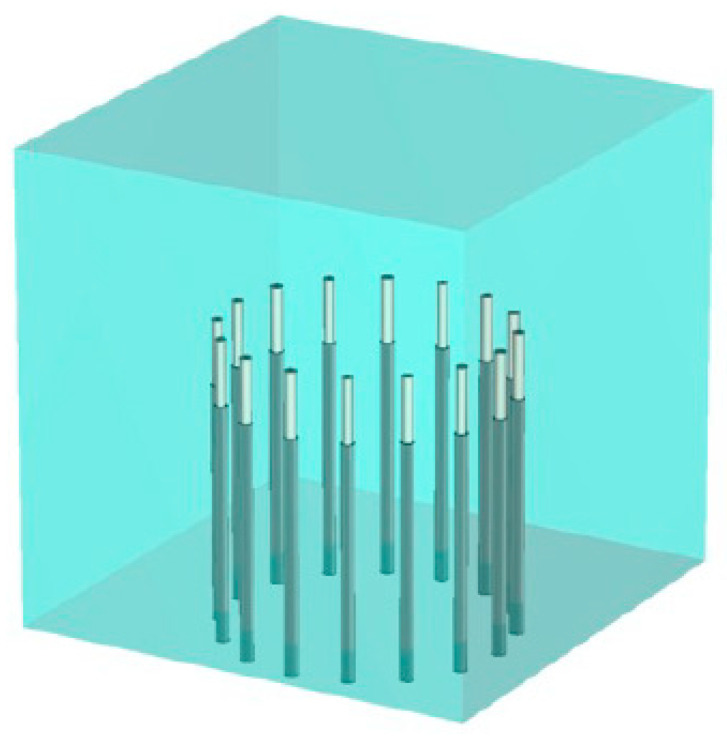
Schematic representation of the full array of 16 monopole antennas.

**Figure 2 sensors-25-05150-f002:**
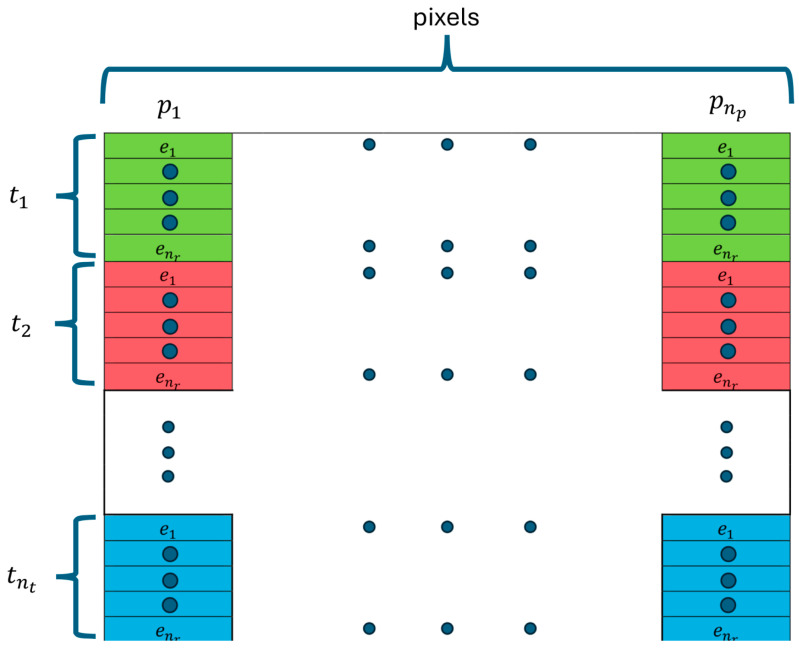
Diagram of the Jacobian matrix used during image reconstruction, describing interactions between image pixels ($p_{1}-\;p_{n_{r}}$), and transmitters ($t_{1}-\;t_{n_{r}}$) and associated receivers ($e_{1}-\;e_{n_{r}}$), respectively.

**Figure 3 sensors-25-05150-f003:**
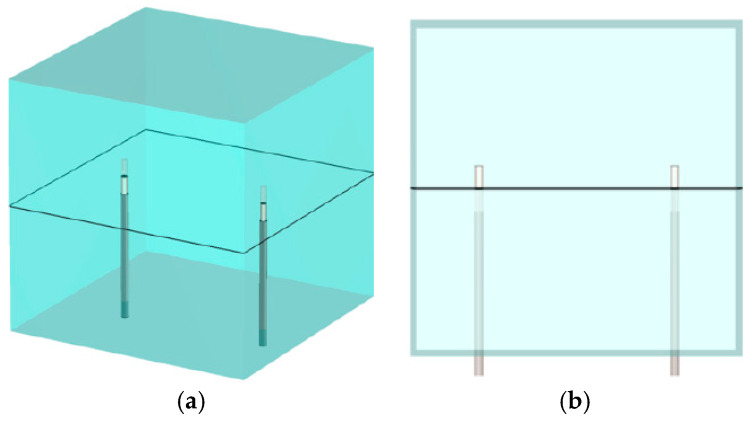
Schematic diagrams of the x–z planes: (**a**) oblique and (**b**) side views, respectively.

**Figure 4 sensors-25-05150-f004:**
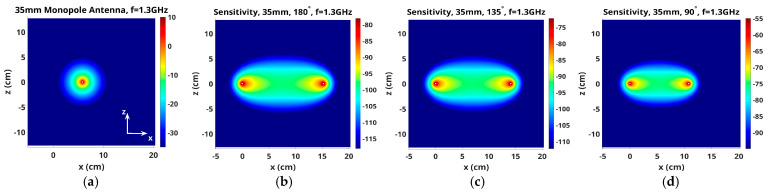
(**a**) Magnitude of electric field in the z–x plane from a 3.5 cm long monopole antenna radiating at 1.3 GHz. Corresponding sensitivity maps for antenna pairs (**b**) 1–9, (**c**) 1–7 and (**d**) 1–5, respectively.

**Figure 5 sensors-25-05150-f005:**
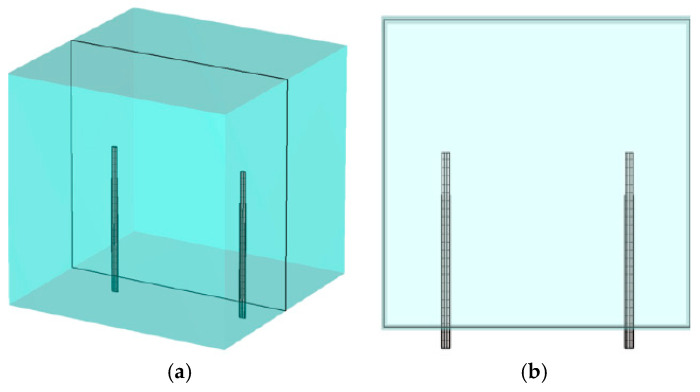
Schematic diagrams of the x–y planes: (**a**) oblique and (**b**) side views, respectively.

**Figure 6 sensors-25-05150-f006:**
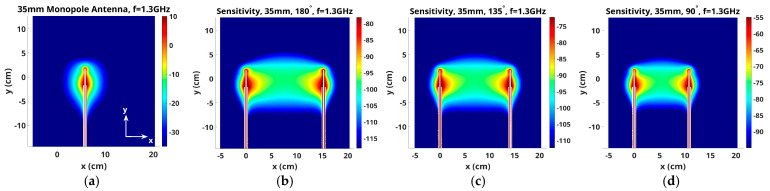
(**a**) Magnitude of the 1.3 GHz field distribution from a 3.5 cm long monopole antenna in the x–y plane. Corresponding sensitivity maps for antenna pairs (**b**) 1–9, (**c**) 1–7 and (**d**) 1–5, respectively.

**Figure 7 sensors-25-05150-f007:**
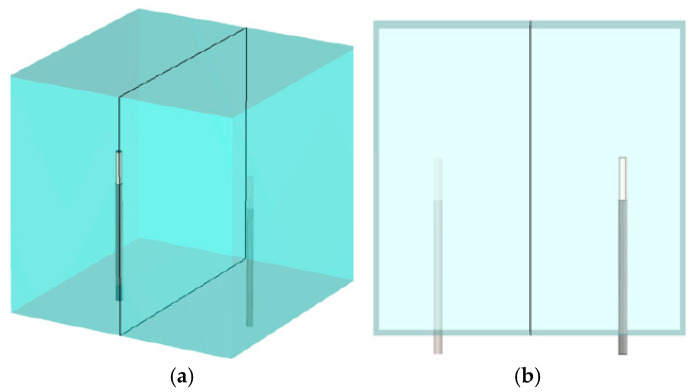
Schematic diagrams of the y–z planes: (**a**) oblique and (**b**) side views, respectively.

**Figure 8 sensors-25-05150-f008:**
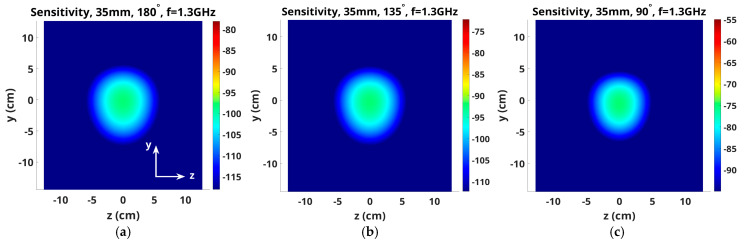
Sensitivity maps in the z–y plane for antenna pairs (**a**) 1–9, (**b**) 1–7 and (**c**) 1–5, respectively.

**Figure 9 sensors-25-05150-f009:**
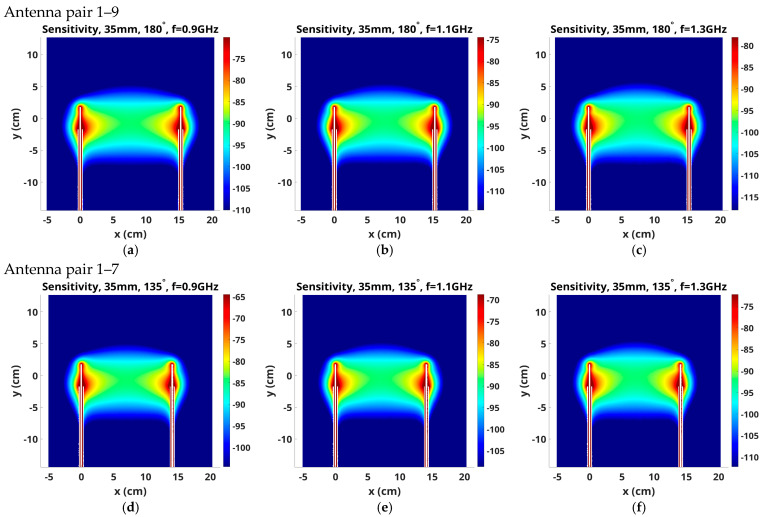
Sensitivity maps in the x–y plane for 3.5 cm long antenna pairs (**a**–**c**) 1–9, (**d**–**f**) 1–7 and (**g**–**i**) 1–5, respectively, radiating at 0.9, 1.1 and 1.3 GHz.

**Figure 10 sensors-25-05150-f010:**
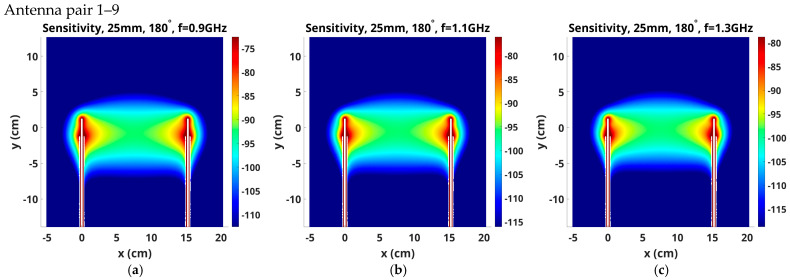
Sensitivity maps in the x-y plane for the 2.5 cm long antenna pairs (**a**–**c**) 1–9, (**d**–**f**) 1–7, and (**g**–**i**) 1–5, respectively, radiating at 0.9, 1.1 and 1.3 GHz.

**Figure 11 sensors-25-05150-f011:**
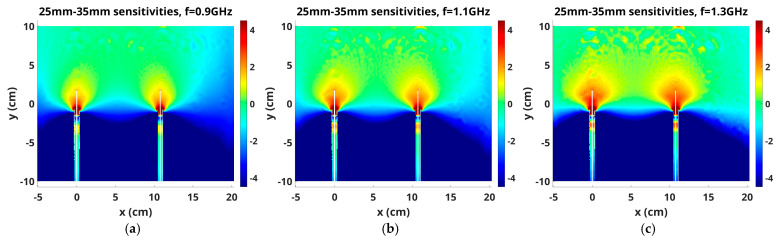
Differences (for the 2.5 cm antenna—the 3.5 cm antenna case) in the sensitivity maps for the 1–9 antenna pair cases for (**a**) 900, (**b**) 1100 and (**c**) 1300 MHz, respectively.

## Data Availability

We are happy to make the data for these simulations available upon request. As we described in this manuscript, the derivations are thoroughly described in preceding publications and their extensions to 3D in this manuscript are straightforward.

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
