# Peer review of "3D Sensitivity Zone Mapping in a Multi-Static, Microwave Breast Imaging Configuration"

_sensors, 2025, doi:10.3390/s25165150_

Round 1

Reviewer 1 Report

Comments and Suggestions for Authors

The paper presents a study on 3D sensitivity zone mapping for monopole antenna arrays in microwave breast imaging. The authors use simulations to visualize the Jacobian matrix for different antenna pairs, frequencies, and antenna lengths. The work extends previous 2D analysis to a more realistic 3D scenario. Overall, the paper is well-written and the results are of high interest. However, there are several points that need to be addressed to improve the manuscript's quality and clarity.

- Could the authors elaborate on the clinical implications of their findings, particularly in terms of improving breast cancer diagnostics?

- The design is appropriate for studying 3D sensitivity zones. The use of monopole antennas and the focus on frequency/length variations are justified. However, the paper lacks a clear hypothesis or research question upfront. The paper would be beneficial from an extra statement of the research objectives or hypotheses at the end of the introduction.

- Figure 2 is intended to illustrate the structure of the Jacobian matrix, but they details it illustrates may cause confusion to reader. I suggest that the authors add a clearer, more intuitive explanation or schematic of the Jacobian method.

- The authors claimed that the sensitivity extending down the coaxial feedlines is not problematic for breast imaging because the tissue tapers away . This is a reasonable assumption for a pendant breast geometry.  However, could this be an issue for other applications or for imaging near the chest wall where the geometry is flatter?

- The number for subsection "Variation on antenna length" is incorrect (should be 3.3 instead of 3.1). Please revise the whole paper and make sure the sections/subsections are numbered with correct order.

Author Response

Reviewer #1

The paper presents a study on 3D sensitivity zone mapping for monopole antenna arrays in microwave breast imaging. The authors use simulations to visualize the Jacobian matrix for different antenna pairs, frequencies, and antenna lengths. The work extends previous 2D analysis to a more realistic 3D scenario. Overall, the paper is well-written and the results are of high interest. However, there are several points that need to be addressed to improve the manuscript's quality and clarity.

- Could the authors elaborate on the clinical implications of their findings, particularly in terms of improving breast cancer diagnostics?

We have modified the Introduction to be more clear about the primary goal of the paper which is to develop a 3D sensitivity tool of assessing the overall design of an imaging array for tomography applications.  Using this analysis, we demonstrate that monopole antennas provide good sensitivity within the primary imaging zone and negligible sensitivity elsewhere – overcoming criticisms of not radiating maximum power into the target of interest and limiting effects from stray scatters outside of the imaging zone.  In addition, because monopoles can be packed close to targets, their sensitivity patterns are less susceptible to gaps in coverage of the target region.  These points have been further emphasized in the revised Introduction and echoed where appropriate in the revised Discussion.

- The design is appropriate for studying 3D sensitivity zones. The use of monopole antennas and the focus on frequency/length variations are justified. However, the paper lacks a clear hypothesis or research question upfront. The paper would be beneficial from an extra statement of the research objectives or hypotheses at the end of the introduction.

We have modified and expanded the Introduction and Discussion sections to indicate that rows in the Jacobian matrix can be viewed as sensitivity maps for the imaging zone and overall transmitter-receiver configuration.  In this sense, these distributions can be used as a tool for overall system design/evaluation. 

- Figure 2 is intended to illustrate the structure of the Jacobian matrix, but the details it illustrates may cause confusion to reader. I suggest that the authors add a clearer, more intuitive explanation or schematic of the Jacobian method.

We added text in the Introduction describing how the Jacobian matrix is expanded from a single variable Newton’s method to its multi-variate form.  We also added a new subsection in the Methods section to describe the expansion explicitly.

- The authors claimed that the sensitivity extending down the coaxial feedlines is not problematic for breast imaging because the tissue tapers away. This is a reasonable assumption for a pendant breast geometry.  However, could this be an issue for other applications or for imaging near the chest wall where the geometry is flatter?

In the imaging configuration evaluated in this paper, the breast is pendant vertically into a space within the antenna array (depicted in Figure 1), and the chestwall is positioned above the antenna tips.  We used the analysis to minimize enhanced sensitivity above the array by selecting longer antennas.  We added text to clarify this point and improve understanding of our feedline arguments.

- The number for subsection "Variation on antenna length" is incorrect (should be 3.3 instead of 3.1). Please revise the whole paper and make sure the sections/subsections are numbered with correct order.

We corrected this error and made sure the numbering change was consistent throughout the rest of the manuscript.

Reviewer 2 Report

Comments and Suggestions for Authors

It is an interesting paper. Based on the given setup consisting 16 monopole antennas, it gives insights about 3D position and channel (antenna pair) dependent sensitivity to changes the electromagnetic properties to be reconstructed. It is well and stringently written. After reading the full text (including discussion and conclusion), it comes clear what consequences the changes in sensitivity practically have or may have.

I found one typo and have a few suggestions for improvement the readability, if there are no objections.

Typo:

  • 9e vs. 9e --> 9e vs. 10e. Furthermore, Figure is not otherwise abbreviated in the text, so it should also be written out here.

Suggestions for improvement:

  • The explanations of the 4 values of the 2x2 sub-matrix as well as the calculation of the sensitivity shown in the figures can be improved. What exactly do the four matrix values indicate? This is difficult to deduce from lines 94-97.
  • According to lines 133-136, the four values are multiplied and then only logarithmized, right? Y could be defined more clearly here using also a short inline formula.
  • The coordinates are unambiguous with the help of the text. If necessary, legibility could be improved by drawing in small coordinate systems or by labeling the axes in figures 3a, 5a and 7a.
  • I am wondering what the reference value is in Figure 4a and 5a. It cannot be the maximum field strength because the dB scale does not start at 0 dB. An additional explanation could help here.
  • From my point of view, compared to the rather smaller bow-like differences, the sensitivity differences between the antennas due to different antenna distances comes up a little short in the description. But maybe, it is clear from the start or it was already discussed in the 2D papers, e.g. [14].
  • The sensitivity maps shown in Figure 9 and 10 are very similar. In order to highlight the differences described in words, I recommend to add few differential sensitivity maps (images of the difference between two relevant sensitivity maps in dB). For example, it is difficult to comprehend the sentence (line 255-257) "The tendency for the upper part of the sensitivity zone to bow upwards with increased frequency and larger antenna spacing appears to be accentuated by the shorter (2.5 cm) antenna lengths." only by comparing images 9c/9g vs. 10c/10g.

Author Response

Reviewer #2

It is an interesting paper. Based on the given setup consisting 16 monopole antennas, it gives insights about 3D position and channel (antenna pair) dependent sensitivity to changes the electromagnetic properties to be reconstructed. It is well and stringently written. After reading the full text (including discussion and conclusion), it comes clear what consequences the changes in sensitivity practically have or may have.

I found one typo and have a few suggestions for improvement the readability, if there are no objections.

Typo:

9e vs. 9e --> 9e vs. 10e. Furthermore, Figure is not otherwise abbreviated in the text, so it should also be written out here.

This typo has been fixed.

Suggestions for improvement:

The explanations of the 4 values of the 2x2 sub-matrix as well as the calculation of the sensitivity shown in the figures can be improved. What exactly do the four matrix values indicate? This is difficult to deduce from lines 94-97.

We expanded this discussion to show these terms explicitly, and indicate their derivations appear in Meaney et al [2001].

According to lines 133-136, the four values are multiplied and then only logarithmized, right? Y could be defined more clearly here using also a short inline formula.

We expanded discussion in the Methods section to first show the partial derivative terms of the 2x2 matrices when transitioned from “real” cases to “complex” cases.  In addition, we show the Jacobian terms explicitly after log transformation (in particular, using the quotient rule).

The coordinates are unambiguous with the help of the text. If necessary, legibility could be improved by drawing in small coordinate systems or by labeling the axes in figures 3a, 5a and 7a.

Small coordinate system axes have been added.

I am wondering what the reference value is in Figure 4a and 5a. It cannot be the maximum field strength because the dB scale does not start at 0 dB. An additional explanation could help here.

As noted in the text, maps of derivative values differ by a scaling factor resulting from the multiplication of two fields together [Hosseinzadegan et al 2021].  Signal levels at coaxial inputs were set to the same level for both antennas in all test cases.  Since division is logarithmic difference between like terms, the absolute value of the source term is subtracted out.  Similarly, the scaling factor required to transform the product of the two field distributions into sensitivity map terms in the Jacobian matrix also gets subtracted out.  We have added this discussion to the revised text.

In addition, the center value of the sensitivity map color scale was defined by the value at the center of the enhanced sensitivity band in each case to facilitate comparisons of sensitivity map characteristics across cases.  Because sensitivity maps are presented for different antenna spacings, antenna lengths and operating frequencies, the associated scalings do vary.

From my point of view, compared to the rather smaller bow-like differences, the sensitivity differences between the antennas due to different antenna distances comes up a little short in the description. But maybe, it is clear from the start or it was already discussed in the 2D papers, e.g. [14]\]

More pronounced differences are likely to occur when in other situations – for example when comparing sensitivity maps of our isotropic, monopole antennas relative to more directive antennas such as waveguides and patch antennas.  However, we were motivated to incorporate actual structures in the imaging system in order to illustrate sensitivity for regions and structures outside of the imaging zone – including feedlines and support structures.  The impact of feedlines is not trivial, and has been mostly ignored.

While the distinguishing effects between two different antenna lengths are less pronounced, they have design implications in our system.  Data from phantom experiments showed images recovered close to the air/liquid interface were corrupted with more artifacts when using the 2.5 cm long antennas compared to when using the 3.5 cm long antennas, and we concluded that the difference was probably due to the antennas themselves.  Differences in the sensitivity maps are subtle, but not insignificant, and confirm our hypothesis.  We now show sensitivity difference maps between the two length cases to emphasize their differences, which indicate a distinct enhancement around and above both antennas that increased substantially with frequency.  Similarly, the region between and above the antenna tips was generally 1-2 dB greater for the shorter antennas and the effect also increased with frequency.  We expanded the text describing these data in the Introduction, Results and Discussion sections.

The sensitivity maps shown in Figure 9 and 10 are very similar. In order to highlight the differences described in words, I recommend to add few differential sensitivity maps (images of the difference between two relevant sensitivity maps in dB). For example, it is difficult to comprehend the sentence (line 255-257) "The tendency for the upper part of the sensitivity zone to bow upwards with increased frequency and larger antenna spacing appears to be accentuated by the shorter (2.5 cm) antenna lengths." only by comparing images 9c/9g vs. 10c/10g.

The reviewer is correct that differences in the sensitivity maps are subtle.  Accordingly, we added a figure showing the difference in sensitivity maps for the two cases.  Two features are notable.  First, a higher elevated zones occurs around and above the ends of the antennas for the 2.5 cm length compared to the 3.5 cm length.  Given the monopole antennas are typically placed within 2 mm of the liquid surface, the zone right near the liquid/air interface could contribute to additional image artifacts.  Second, an elevated difference area occurs between and above the tips of the antennas, which could also contribute to unwanted signal interactions with the liquid/air interface.